# QJL: 1-Bit Quantized JL Transform for KV Cache Quantization with Zero Overhead

## Abstract

Serving LLMs requires substantial memory due to the storage requirements of Key-Value (KV) embeddings in the KV cache, which grows with sequence length. An effective approach to compress KV cache is quantization. However, traditional quantization methods face significant memory overhead due to the need to store quantization constants (at least a zero point and a scale) in full precision per data block. Depending on the block size, this overhead can add 1 or 2 bits per quantized number. We introduce QJL, a new quantization approach that consists of a Johnson-Lindenstrauss (JL) transform followed by sign-bit quantization. In contrast to existing methods, QJL eliminates memory overheads by removing the need for storing quantization constants. We propose an asymmetric estimator for the inner product of two vectors and demonstrate that applying QJL to one vector and a standard JL transform without quantization to the other provides an unbiased estimator with minimal distortion. We have developed an efficient implementation of the QJL sketch and its corresponding inner product estimator, incorporating a lightweight CUDA kernel for optimized computation. When applied across various LLMs and NLP tasks to quantize the KV cache to only 3 bits, QJL demonstrates a more than fivefold reduction in KV cache memory usage without compromising accuracy, all while achieving faster runtime.

## 1 Introduction

Large language models (LLMs) have garnered significant attention and demonstrated remarkable success in recent years. Their applications span various domains, including chatbot systems Achiam et al. (2023); Antropic (2024) to text-to-image Ramesh et al. (2022); FireFly (2023); Midjourney (2022), text-to-video synthesis OpenAI (2024b), coding assistant Copilot (2023) and even multimodal domain across text, audio, image, and video OpenAI (2024a). The Transformer architecture with self-attention mechanism Vaswani et al. (2017) is at the heart of these LLMs as it enables capturing intrinsic pairwise correlations across tokens in the input sequence. The ability of LLMs grows along with their model size Kaplan et al. (2020), which leads to computational challenges in terms of huge memory consumption.

Deploying auto-regressive transformers during the generation phase is costly because commercial AI models must simultaneously serve millions of end users while meeting strict latency requirements. One significant challenge is the substantial memory needed to store all previously generated key-value (KV) embeddings in cache to avoid recomputations. This has become a major memory and speed bottleneck, especially for long context lengths. Additionally, the GPU must load the entire KV cache from its main memory to shared memory for each token generated, resulting in low arithmetic intensity and leaving most GPU threads idle. Therefore, reducing the KV cache size while maintaining accuracy is crucial.

There are several approaches to address this challenge. One method involves reducing the number of heads in the KV cache using multi-query attention Shazeer (2019) and multi-group attention Ainslie et al. (2023), but these require fine-tuning the pre-trained models or training from scratch. Another line of work tries to reduce the KV cache size by pruning or evicting unimportant tokens Zhang et al. (2024b); Liu et al. (2024a); Xiao et al. (2023); Zandieh et al. (2024). Additionally, some recent works tackle the issue from a system perspective, such as offloading Sheng et al. (2023) or using virtual memory and paging techniques in the attention mechanism Kwon et al. (2023).

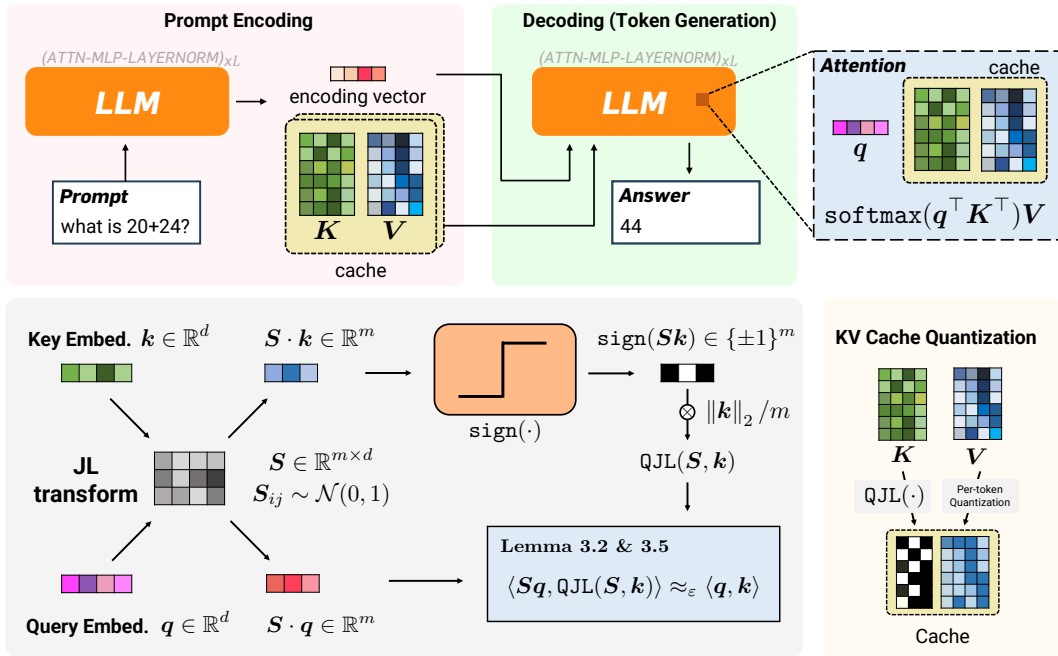

Figure 1: Overview of the KV cache quantization via Quantized JL (QJL) transform

A simple yet effective approach is to quantize the floating-point numbers (FPN) in the KV cache using fewer bits. Several quantization methods have been proposed specifically for the KV cache Yue et al. (2024); Yang et al. (2024); Dong et al. (2024); Kang et al. (2024); Zhang et al. (2024a). Most recently, KIVI Liu et al. (2024b) and KVQuant Hooper et al. (2024) proposed per-channel quantization for the key cache to achieve better performance. However, all existing quantization methods for the KV cache face significant "memory overhead" issues. Specifically, all these methods group the data into blocks, either channel-wise or token-wise, and calculate and store quantization constants (at least a zero point and a scale) for each group. Depending on the group size, this overhead can add approximately 1 or 2 additional bits per quantized number, which results in significant computational overhead. In this work, our goal is to develop an efficient, data-oblivious quantization method, referred to as a *sketching technique*. This method, which we call QJL, does not need to be tuned by or adapted to the input data with significantly less overhead than prior works, without any loss in performance.

## 1.1 OVERVIEW OF CONTRIBUTIONS

The decoding phase in the attention mechanism involves the following computations: (1) computing attention scores by applying the softmax function to the inner product between the current query embedding and all previously generated keys, and (2) multiplying the attention scores with all previously generated values. To make the attention score calculations in step (1) more memory efficient, we quantize the keys in the cache. We introduce a quantization scheme for key embeddings, named QJL, leveraging randomized sketching techniques. Alongside, we develop a high-accuracy estimator for the inner product of query/key pairs, crucial for mitigating errors amplified by the softmax operation in attention score calculations.

Firstly, we revisit a fundamental concept in numerical linear algebra: applying a Johnson-Lindenstrauss (JL) transform, i.e., a random Gaussian projection, to a pair of vectors and then computing the inner product of the projected vectors provides an unbiased and low-distortion estimator for their original inner product Dasgupta & Gupta (2003). To address the key cache quantization problem, our aim is to quantize the result after applying the JL transform to a key embedding, ideally to just a single bit. Surprisingly, we prove that by applying the JL transform to a key embedding and then quantizing the result to a single bit (the sign bit), while applying the same JL transform to the

query embedding without quantization, we still obtain an unbiased estimator of their inner product (see Lemma 3.2). Moreover, the distortion of this estimator is small and comparable to that of the standard JL transform (see Lemma 3.5). In Theorem 3.6, we demonstrate that the proposed inner product estimator based on QJL achieves a relative distortion of $1 \pm \varepsilon$ on the final attention scores. Notably, the number of required bits for representing quantized keys is independent of the embedding dimension and scales logarithmically with the context length, using a fixed number of bits per token.

Thus the QJL sketch combines a JL transform—a random Gaussian projection—with quantization to the sign bit. An overview of this approach is illustrated in Figure 1. Unlike previous methods, the QJL sketch can quantize vectors with zero overhead because it does not require grouping the data and storing quantization constants (zeros and scales) per group. Furthermore, this is a data-oblivious algorithm that does not rely on specific input, requires no tuning, and can be easily parallelized and applied in real-time.

The value cache quantization used to make step (2) memory efficient is known to be a straightforward task, and a standard token-wise quantization is very effective and efficient in practice, as observed in prior work Liu et al. (2024b); Hooper et al. (2024). Hence, we follow the same approach for the value therein.

Furthermore, we analyzed the distribution of outliers in large language models (LLMs). We observed that while there are no significant outliers in the initial layers, certain fixed key embedding channels (coordinates) in the deeper layers exhibit considerably larger magnitudes (see Figure 2). To address this, we identify these outlier channels during the prompt phase and simply apply two independent copies of our quantizer to the outliers and inliers separately.

The QJL transform and its accompanying inner product estimator are highly efficient and GPU-friendly algorithms. In particular, we provide a lightweight CUDA kernel for their efficient computation. We apply QJL and our inner product estimator to compress the KV cache in several LLMs, including Llama-2 Touvron et al. (2023) and its fine-tuned models by long sequence Li et al. (2023), under various NLP tasks. Our results show that quantizing the KV cache to only 3 bits per FPN results in no accuracy drop compared to the exact model with 16 bits per FPN while reducing cache memory usage by over fivefold and increasing the generation speed significantly for long contexts. For example, our proposed quantization shows better F1 scores on long-range question-answering tasks from LongBench Bai et al. (2023) (a collection of long-context datasets) compared to the recent KV cache quantization methods, while minimizing memory overheads.

## 2 PRELIMINARIES: TOKEN GENERATION IN ATTENTION

Deploying auto-regressive language models for inference involves performing attention decoding in an online setting, where key and value embeddings from each transformer layer are cached in memory to remove redundant computations. The model sequentially uses and updates the KV cache to generate the next token, one at a time.

More precisely, in every phase of token generation, the stream of tokens is represented by a triplet of vectors called by the query, key, and value embeddings, respectively. Let $\boldsymbol{q}_i, \boldsymbol{k}_i, \boldsymbol{v}_i \in \mathbb{R}^d$ be the triplet at $i$-th generation phase and $n$ be the total number of tokens in the stream so far either in the prompt encoding (prefill) or the generation (decoding) phase. Then, the attention output in $n$-th generation phase can be written as

$$\boldsymbol{o}_n = \sum_{i \in [n]} \texttt{Score}(i) \cdot \boldsymbol{v}_i, \tag{1}$$

where $\texttt{Score} \in \mathbb{R}^n$ is the vector of attention scores defined as:

$$\texttt{Score} := \texttt{softmax}\left([\langle \boldsymbol{q}_n, \boldsymbol{k}_1 \rangle, \langle \boldsymbol{q}_n, \boldsymbol{k}_2 \rangle, \dots \langle \boldsymbol{q}_n, \boldsymbol{k}_n \rangle]\right). \tag{2}$$

The output embedding $\boldsymbol{o}_n$ will be used for computing the next tokens in the stream $\boldsymbol{q}_{n+1}, \boldsymbol{k}_{n+1}, \boldsymbol{v}_{n+1}$ unless the generation phase terminates. Observe that to compute output $\boldsymbol{o}_n$, one needs to store all previous key and value embeddings $\{\boldsymbol{k}_i, \boldsymbol{v}_i\}_{i \in [n]}$ and keeping them in full precision requires significant memory for long-context inputs. The time complexity to compete Equation (2) is $O(nd)$ due to the computation of $n$ inner products. Additionally, the inference speed is also impacted by the KV cache size, as the KV cache must be loaded from GPU main memory for every token generated,

resulting in low arithmetic intensity and underutilization of GPU cores Pope et al. (2023). In this work, we focus on compressing the KV cache by quantizing tokens, thereby reducing the memory required to store each key or value embedding in the cache.

## 3 QUANTIZED JOHNSON-LINDENSTRAUSS (QJL) TRANSFORM

Our goal is to save memory space for storing the KV cache while the inner product between query and key remains undistorted. To achieve this, we first transform the embedding vectors using a random projection that preserves the inner products, acting as a preconditioning step, and then quantize the result. Specifically, we project the input vectors onto a random subspace by applying the Johnson-Lindenstrauss (JL) transform Johnson et al., which amounts to multiplying by a random Gaussian matrix. The inner product of the resulting vectors after applying this projection provides an unbiased and low-distortion estimator for the inner product of the original vectors Dasgupta & Gupta (2003). We introduce a 1-bit Johnson-Lindenstrauss transform, comprising a JL transformation followed by quantization to a single sign bit, and demonstrate its ability to offer an unbiased and low-distortion inner product estimator. We complement our binary quantizer by developing an unbiased estimator for the inner product of the quantized vector with any arbitrary vector. This inner product estimator is asymmetric, as one of the vectors is quantized to a single bit while the other remains unquantized, making it well-suited for the KV cache mechanism. The Quantized Johnson-Lindenstrauss (QJL) transformation, acting as a 1-bit quantizer, alongside our proposed estimator, is formally defined in the following definition:

**Definition 3.1** (QJL and inner product estimator). For any positive integers $d, m$, let $\boldsymbol{S} \in \mathbb{R}^{m \times d}$ be a JL transform matrix, i.e., entries of $\boldsymbol{S}$ are i.i.d. samples from the zero mean and unit variance Normal distribution. The QJL is a mapping function $\mathcal{H}_S : \mathbb{R}^d \to \{-1, +1\}^m$ defined as:

$$\mathcal{H}_S(\boldsymbol{k}) := \texttt{sign}(\boldsymbol{S}\boldsymbol{k}) \quad \text{for any } \boldsymbol{k} \in \mathbb{R}^d. \tag{3}$$

Furthermore, for any pair of vectors $\boldsymbol{k}, \boldsymbol{q} \in \mathbb{R}^d$ the estimator for their inner product $\langle \boldsymbol{q}, \boldsymbol{k} \rangle$ based on the aforementioned quantizer is defined as:

$$\texttt{Prod}_{\texttt{QJL}}(\boldsymbol{q}, \boldsymbol{k}) := \frac{\sqrt{\pi/2}}{m} \cdot \|\boldsymbol{k}\|_2 \cdot \langle \boldsymbol{S}\boldsymbol{q}, \mathcal{H}_S(\boldsymbol{k}) \rangle. \tag{4}$$

Now, we show that the inner product estimator $\texttt{Prod}_{\texttt{QJL}}(\boldsymbol{q}, \boldsymbol{k})$, exactly like the inner product of JL-transformed vectors without quantization to sign bit, is an unbiased estimator. The crucial point to note is that if we applied QJL to both vectors $\boldsymbol{q}$ and $\boldsymbol{k}$ in Equation (4), we would obtain an unbiased estimator for the angle between these vectors, as shown in Charikar (2002). However, to estimate the inner product one needs to apply the cosine function on top of the angle estimator, which results in a biased estimation. Thus, to achieve an unbiased inner product estimator, it is necessary to asymmetrically apply quantization to the JL transform of only one of the vectors $\boldsymbol{q}$ and $\boldsymbol{k}$.

**Lemma 3.2** (Inner product estimator $\texttt{Prod}_{\texttt{QJL}}$ is unbiased). *For any vectors $\boldsymbol{q}, \boldsymbol{k} \in \mathbb{R}^d$ the expected value of the estimator $\texttt{Prod}_{\texttt{QJL}}(\boldsymbol{q}, \boldsymbol{k})$ defined in Equation* (4) *is:*

$$\mathbb{E}_{\boldsymbol{S}}[\texttt{Prod}_{\texttt{QJL}}(\boldsymbol{q}, \boldsymbol{k})] = \langle \boldsymbol{q}, \boldsymbol{k} \rangle,$$

*where the expectation is over the randomness of the JL matrix $\boldsymbol{S}$ in Definition 3.1.*

*Proof.* Let $\boldsymbol{s}_1, \boldsymbol{s}_2, \ldots \boldsymbol{s}_m$ denote the rows of the JL matrix $\boldsymbol{S}$. Additionally, let us decompose $\boldsymbol{q}$ to its projection onto the vector $\boldsymbol{k}$ and its orthogonal component, i.e., $\boldsymbol{q}^{\perp k} := \boldsymbol{q} - \frac{\langle \boldsymbol{q}, \boldsymbol{k} \rangle}{\|\boldsymbol{k}\|_2^2} \cdot \boldsymbol{k}$. We can write,

$$
\begin{aligned}
\texttt{Prod}_{\texttt{QJL}}(\boldsymbol{q}, \boldsymbol{k}) &= \frac{\sqrt{\pi/2}}{m} \sum_{i \in [m]} \|\boldsymbol{k}\|_2 \cdot \boldsymbol{s}_i^\top \boldsymbol{q} \cdot \texttt{sign}(\boldsymbol{s}_i^\top \boldsymbol{k}) \\
&= \frac{\sqrt{\pi/2}}{m} \sum_{i \in [m]} \frac{\langle \boldsymbol{q}, \boldsymbol{k} \rangle}{\|\boldsymbol{k}\|_2} \cdot \boldsymbol{s}_i^\top \boldsymbol{k} \cdot \texttt{sign}(\boldsymbol{s}_i^\top \boldsymbol{k}) + \|\boldsymbol{k}\|_2 \cdot \boldsymbol{s}_i^\top \boldsymbol{q}^{\perp k} \cdot \texttt{sign}(\boldsymbol{s}_i^\top \boldsymbol{k}) \\
&= \frac{\sqrt{\pi/2}}{m} \sum_{i \in [m]} \frac{\langle \boldsymbol{q}, \boldsymbol{k} \rangle}{\|\boldsymbol{k}\|_2} \cdot |\boldsymbol{s}_i^\top \boldsymbol{k}| + \|\boldsymbol{k}\|_2 \cdot \boldsymbol{s}_i^\top \boldsymbol{q}^{\perp k} \cdot \texttt{sign}(\boldsymbol{s}_i^\top \boldsymbol{k}).
\end{aligned}
$$

Since $\boldsymbol{s}_i$'s have identical distributions, we have:

$$\underset{\boldsymbol{S}}{\mathbb{E}}[\texttt{Prod}_{\texttt{QJL}}(\boldsymbol{q}, \boldsymbol{k})] = \sqrt{\pi/2} \left( \frac{\langle \boldsymbol{q}, \boldsymbol{k} \rangle}{\|\boldsymbol{k}\|_2} \cdot \mathbb{E}\left[|\boldsymbol{s}_1^\top \boldsymbol{k}|\right] + \|\boldsymbol{k}\|_2 \cdot \mathbb{E}\left[\boldsymbol{s}_1^\top \boldsymbol{q}^{\perp k} \cdot \texttt{sign}(\boldsymbol{s}_1^\top \boldsymbol{k})\right] \right).$$

To calculate the above expectation let us define variables $x := \boldsymbol{s}_1^\top \boldsymbol{k}$ and $y := \boldsymbol{s}_1^\top \boldsymbol{q}^{\perp k}$. Note that $x$ and $y$ are both zero-mean Gaussian random variables and because $\langle \boldsymbol{q}^{\perp k}, \boldsymbol{k} \rangle = 0$. By the following Fact 3.3, $x$ and $y$ are independent.

*Fact* 3.3. If $\boldsymbol{x} \in \mathbb{R}^d$ is a vector of i.i.d. zero-mean normal entries with variance $\sigma^2$ and $A \in \mathbb{R}^{m \times d}$ is a matrix, then $\boldsymbol{A} \cdot \boldsymbol{x}$ is a normal random variable with mean zero and covariance matrix $\sigma^2 \cdot \boldsymbol{A}\boldsymbol{A}^\top$.

This implies that the second expectation term above is zero because $\mathbb{E}\left[\boldsymbol{s}_1^\top \boldsymbol{q}^{\perp k} \cdot \texttt{sign}(\boldsymbol{s}_1^\top \boldsymbol{k})\right] = \mathbb{E}[y \cdot \texttt{sign}(x)] = \mathbb{E}[y] \cdot \mathbb{E}[\texttt{sign}(x)] = 0$. Furthermore, $x$ is a Gaussian random variable with mean zero and variance $\|\boldsymbol{k}\|_2^2$. Therefore, we have

$$\underset{\boldsymbol{S}}{\mathbb{E}}[\texttt{Prod}_{\texttt{QJL}}(\boldsymbol{q}, \boldsymbol{k})] = \sqrt{\pi/2} \cdot \frac{\langle \boldsymbol{q}, \boldsymbol{k} \rangle}{\|\boldsymbol{k}\|_2} \cdot \underset{x}{\mathbb{E}}[|x|] = \langle \boldsymbol{q}, \boldsymbol{k} \rangle.$$

where the equality comes from the following Fact 3.4:

*Fact* 3.4 (Moments of Normal Random Variable). If $x$ is a normal random variable with zero mean and variance $\sigma^2$, then for any integer $\ell$, the $\ell$-th moment of $x$ is $\mathbb{E}\left[|x|^\ell\right] = \sigma^\ell \cdot 2^{\ell/2}\Gamma((\ell+1)/2)/\sqrt{\pi}$.

This completes the proof of Lemma 3.2. $\qquad\square$

Now we show that the inner product estimator $\texttt{Prod}_{\texttt{QJL}}$ in Definition 3.1, just like the estimators based on the standard JL transform, has a bounded distortion with high probability.

**Lemma 3.5** (Distortion of inner product estimator $\texttt{Prod}_{\texttt{QJL}}$). *For any vectors $\boldsymbol{q}, \boldsymbol{k} \in \mathbb{R}^d$ if the estimator $\texttt{Prod}_{\texttt{QJL}}(\boldsymbol{q}, \boldsymbol{k})$ is defined as in Equation* (4) *for QJL with dimension $m \geq \frac{4}{3} \cdot \frac{1+\varepsilon}{\varepsilon^2} \log \frac{2}{\delta}$, then:*

$$\underset{\boldsymbol{S}}{\Pr}\left[|\texttt{Prod}_{\texttt{QJL}}(\boldsymbol{q}, \boldsymbol{k}) - \langle \boldsymbol{q}, \boldsymbol{k} \rangle| > \varepsilon \|\boldsymbol{q}\|_2 \|\boldsymbol{k}\|_2\right] \leq \delta,$$

*where the probability is over the randomness of the JL matrix $\boldsymbol{S}$ in Definition 3.1.*

*Proof.* First note that, letting $\boldsymbol{s}_1, \boldsymbol{s}_2, \ldots \boldsymbol{s}_m$ denote the rows of the JL transform matrix $S$, we have:

$$\texttt{Prod}_{\texttt{QJL}}(\boldsymbol{q}, \boldsymbol{k}) = \frac{1}{m} \sum_{i \in [m]} \sqrt{\pi/2} \cdot \|\boldsymbol{k}\|_2 \cdot \boldsymbol{s}_i^\top \boldsymbol{q} \cdot \texttt{sign}(\boldsymbol{s}_i^\top \boldsymbol{k}).$$

Since $\boldsymbol{s}_i$'s are i.i.d. the above is indeed the average of $m$ i.i.d. estimators defined as $z_i := \sqrt{\pi/2} \cdot \|\boldsymbol{k}\|_2 \cdot \boldsymbol{s}_i^\top \boldsymbol{q} \cdot \texttt{sign}(\boldsymbol{s}_i^\top \boldsymbol{k})$ for $i \in [m]$. Let us now calculate the $\ell$-th moment of $z_i$ using Fact 3.4:

$$\mathbb{E}\left[|z_i|^\ell\right] = \left(\sqrt{\pi/2} \cdot \|\boldsymbol{k}\|_2\right)^\ell \cdot \mathbb{E}\left[|\boldsymbol{s}_i^\top \boldsymbol{q}|^\ell\right] = \left(\sqrt{\pi} \cdot \|\boldsymbol{k}\|_2 \|\boldsymbol{q}\|_2\right)^\ell \cdot \frac{\Gamma((\ell+1)/2)}{\sqrt{\pi}}, \qquad (5)$$

where the second equality above follows because $\boldsymbol{s}_i^\top \boldsymbol{q}$ is a Gaussian random variable with mean zero and variance $\|\boldsymbol{q}\|_2^2$ along with Fact 3.4. Now we can prove the result by invoking the unbiasedness of the estimator, Lemma 3.2, along with an appropriate version of Bernstein inequality and using the moment bounds in Equation (5). More specifically, our moment calculation in Equation (5) implies:

$$\mathbb{E}\left[|z_i|^\ell\right] = \mathbb{E}\left[|z_i|^2\right] \cdot \left(\sqrt{\pi}\|\boldsymbol{k}\|_2\|\boldsymbol{q}\|_2\right)^{\ell-2} \cdot \frac{\Gamma((\ell+1)/2)}{\Gamma(3/2)} \leq \mathbb{E}\left[|z_i|^2\right] \cdot \left(\frac{2}{3} \cdot \|\boldsymbol{k}\|_2\|\boldsymbol{q}\|_2\right)^{\ell-2} \cdot \frac{\ell!}{2}$$

Therefore, by invoking a proper version of the Bernstein inequality, for instance Corollary 2.11 from Boucheron et al. (2003), we have the following:

$$\underset{\boldsymbol{S}}{\Pr}\left[|\texttt{Prod}_{\texttt{QJL}}(\boldsymbol{q}, \boldsymbol{k}) - \langle \boldsymbol{q}, \boldsymbol{k} \rangle| > t\right] \leq 2 \exp\left(\frac{3}{4} \cdot \frac{mt^2}{\|\boldsymbol{k}\|_2^2\|\boldsymbol{q}\|_2^2 + \|\boldsymbol{k}\|_2\|\boldsymbol{q}\|_2 \cdot t}\right).$$

If we set $t = \varepsilon \|\boldsymbol{q}\|_2 \|\boldsymbol{k}\|_2$ the above simplifies to:

$$\underset{\boldsymbol{S}}{\Pr}\left[|\texttt{Prod}_{\texttt{QJL}}(\boldsymbol{q}, \boldsymbol{k}) - \langle \boldsymbol{q}, \boldsymbol{k} \rangle| > \varepsilon \|\boldsymbol{q}\|_2 \|\boldsymbol{k}\|_2\right] \leq 2 \exp\left(\frac{3}{4} \cdot \frac{m\varepsilon^2}{1+\varepsilon}\right).$$

Therefore if $m \geq \frac{4}{3} \cdot \frac{1+\varepsilon}{\varepsilon^2} \log \frac{2}{\delta}$ the error bound follows. This completes the proof of Lemma 3.5. $\quad\square$

---

**Algorithm 1** QJL Key Cache Quantizer

---

**Input:** Stream of key tokens $\boldsymbol{k}_1, \boldsymbol{k}_2, \ldots \in \mathbb{R}^d$, integer $m$
1: Draw a random sketch $\boldsymbol{S} \in \mathbb{R}^{m \times d}$ with i.i.d. entries $\boldsymbol{S}_{i,j} \sim \mathcal{N}(0, 1)$ as per Definition 3.1
2: **repeat**
3:      Compute $\tilde{\boldsymbol{k}}_i \leftarrow \text{sign}\left(\boldsymbol{S}\boldsymbol{k}_i\right)$ and $\nu_i \leftarrow \|\boldsymbol{k}_i\|_2$
4:      **store** the quantized vector $\tilde{k}_i$ and the key norm $\nu_i$ in the cache
5: **until** token stream ends

---

**Procedure** ESTIMATESCORES($\boldsymbol{q}_n$)

6: Compute inner product estimators $\widetilde{\mathbf{qK}}(j) \leftarrow \frac{\sqrt{\pi/2}}{m} \cdot \nu_i \cdot \langle \boldsymbol{S}\boldsymbol{q}_n, \tilde{\boldsymbol{k}}_j \rangle$ for every $j \in [n]$
7: $\widetilde{\text{Score}} \leftarrow \text{softmax}\left(\widetilde{\mathbf{qK}}\right)$

**return** $\widetilde{\text{Score}}$

---

Note that the distortion bound in Lemma 3.5 has remarkably small constants, even smaller than those of the original unquintized JL transform. This indicates that quantizing one of the vectors to just a single sign bit does not result in any loss of accuracy. We use these properties of QJL and our inner product estimator to prove the final approximation bound on our KV cache quantizer.

## 3.1 KEY CACHE QUANTIZATION VIA QJL

The key cache is used in the computation of attention scores as shown in Equation (2). To calculate these scores, we need to compute the inner products of the current query embedding with all key embeddings in the cache. We design a quantization scheme that allows for a low-distortion estimate of the inner products between an arbitrary query and all keys in the cache. In this section, we develop a practical algorithm with provable guarantees based on QJL and the inner product estimator defined in Definition 3.1.

The quantization scheme presented in Algorithm 1 applies QJL, defined in Definition 3.1, to each key embedding, mapping them to binary vectors and storing the results in the key cache. We show in the following theorem that the attention scores calculated by Algorithm 1 have very small $(1 \pm \varepsilon)$ relative distortion with high probability:

**Theorem 3.6** (Distortion bound on QJL key cache quantizer)**.** *For any sequence of key tokens $\boldsymbol{k}_1, \ldots \boldsymbol{k}_n \in \mathbf{R}^d$ and any integer $m$, Algorithm 1 stores binary vectors $\tilde{\boldsymbol{k}}_1, \ldots \tilde{\boldsymbol{k}}_n \in \{-1, +1\}^m$ along with scalar values $\nu_1, \ldots \nu_n$ in the cache. If the key embeddings have bounded norm $\max_{i \in [n]} \|\boldsymbol{k}_i\|_2 \le r$ and $m \ge 2r^2 \varepsilon^{-2} \log n$, then for any query embedding $\boldsymbol{q}_n \in \mathbf{R}^d$ with bounded norm $\|\boldsymbol{q}_n\|_2 \le r$ the output of the procedure ESTIMATESCORES($\boldsymbol{q}_n$) satisfies the following with probability $1 - \frac{1}{\text{poly}(n)}$ simultaneously for all $i \in [n]$:*

$$\left|\widetilde{\text{Score}}(i) - \text{Score}(i)\right| \le 3\varepsilon \cdot \text{Score}(i),$$

*where* Score *is the vector of attention scores defined in Equation* (2)*.*

*Proof.* The proof is by invoking Lemma 3.5 and a union bound. For every $j \in [n]$ the estimator $\widetilde{\mathbf{qK}}(j)$ computed in line 6 of Algorithm 1 is in fact equal to the inner product estimator $\widetilde{\mathbf{qK}}(j) = \text{Prod}_{\text{QJL}}(\boldsymbol{q}_n, \boldsymbol{k}_j)$ as defined in Equation (4). Thus by Lemma 3.5 we have the following with probability at least $1 - \frac{1}{n^{3/(2+2\varepsilon)}}$:

$$\left|\widetilde{\mathbf{qK}}(j) - \langle \boldsymbol{q}_n, \boldsymbol{k}_j \rangle\right| \le \frac{\varepsilon}{r^2} \cdot \|\boldsymbol{q}_n\|_2 \|\boldsymbol{k}_j\|_2 \le \varepsilon,$$

where the second inequality follows from the preconditions of the theorem regarding the boundedness of the norms of the query and key embeddings. By union bound, the above inequality holds simultaneously for all $j \in [n]$ with high probability in $n$. Thus after applying the softmax function in line 7 of Algorithm 1 we get that with high probability in $n$:

$$\widetilde{\text{Score}}(i) \in e^{\pm 2\varepsilon} \cdot \text{Score}(i) \in (1 \pm 3\varepsilon) \cdot \text{Score}(i).$$

This completes the proof of Theorem 3.6. $\qquad\square$

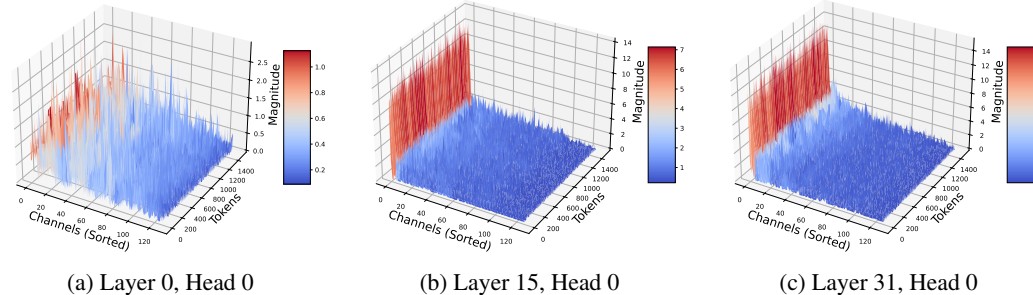

(a) Layer 0, Head 0          (b) Layer 15, Head 0          (c) Layer 31, Head 0

Figure 2: The magnitude of key cache entries for different layers of the Llama-2 model, based on an example prompt, reveals notable patterns. The coordinates of embeddings (channels) are sorted by their average magnitude over tokens. In the initial layers, no significant outlier patterns are observed. However, in the deeper layers, a few channels (approximately four) exhibit visibly larger magnitudes, indicating the presence of significant outliers. This observation highlights the importance of addressing these outliers to improve quantization accuracy and reduce distortion in the key cache.

This theorem shows that if the query and key embeddings have constant norms, as is common in practical scenarios, we can quantize each key embedding such that only $m \approx \varepsilon^{-2} \log n$ bits are needed to store each key token. This is independent of the embedding dimension of the tokens and scales only logarithmically with the sequence length.

## 3.2 VALUE CACHE QUANTIZATION

We quantize the value cache using a standard quantization method, i.e., normalizing each token's entries and then rounding each entry to a few-bit integer representation. This approach aligns with prior work, which has shown that standard token-wise quantization is highly effective for the value cache and results in a minimal accuracy drop Liu et al. (2024b); Hooper et al. (2024).

## 4 EXPERIMENTS

In this section, we validate the empirical performance of our algorithm. All experiments are conducted under a single A100 GPU with 80GB memory. We implement two main CUDA kernels for our core primitives: one for quantizing embedding vectors using various floating point data types such as bfloat16, FP16, and FP32, and the other for computing the inner product of an arbitrary embedding vector with all quantized vectors in the cache. The algorithm's wrapper is implemented in PyTorch, handling all the housekeeping tasks. We plan to complete implementation in the CUDA for future work, which will further accelerate our algorithm.

### 4.1 PRACTICAL CONSIDERATION

**Outliers.** As reported in recent works e.g., KIVI Liu et al. (2024b), KVQuant Hooper et al. (2024), key embeddings typically contain outliers exhibiting a distinct pattern. Specifically, certain coordinates of key embeddings display relatively large magnitudes. To further investigate these observations, we analyze the distribution of the magnitudes of key embedding coordinates across different layers. Firstly, we observe that there are no significant outliers in the initial attention layers. However, in the deeper layers, certain fixed coordinates of key embeddings consistently exhibit large magnitudes, and this pattern persists within these channels across all tokens. The distribution of outliers across different layers for the Llama-2 model is plotted in Figure 2. It is evident that in the initial layers, outliers are rare, but as we approach the final layers, their frequency and impact increase significantly. Secondly, the outliers show a persistent pattern in specific fixed coordinates of the key embeddings. This observation aligns with previous findings that certain fixed embedding coordinates exhibit larger outliers Dettmers et al. (2022); Lin et al. (2023); Liu et al. (2024b); Hooper et al. (2024).

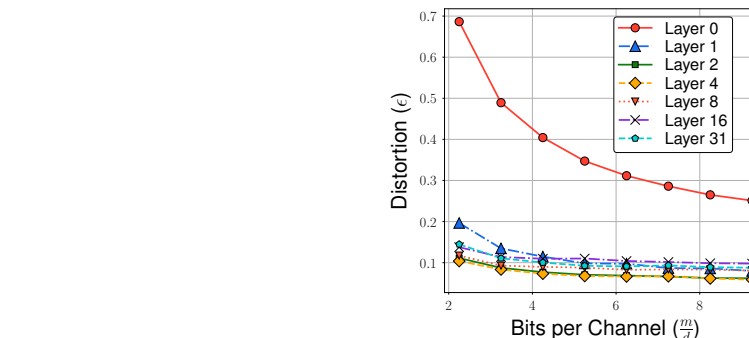

Figure 3: The relative distortion on the attention scores $\varepsilon$ versus the number of bits of QJL per token and embedding channels, i.e., $m/d$, for layers at different depths of Llama 2 model.

As demonstrated in Theorem 3.6, the distortion on the attention scores is directly proportional to the norms of the embeddings. Therefore, capturing these outlier coordinates is essential, as their large magnitudes contribute significantly to the norms of key embeddings. By identifying and isolating these outlier channels, we can reduce the norm of the key embeddings and, consequently, significantly decrease the final distortion. Next, we quantize the outliers using an independent instance of our QJL quantizer but with a lower compression rate, utilizing more bits to accurately represent each outlier coordinate.

**Orthogonalized JL transform.** We observed that orthogonalizing the rows of the JL matrix $S$ in Definition 3.1 almost always improves the performance of our QJL quantizer. This finding aligns with previous work on various applications of the JL transform, such as random Fourier features Yu et al. (2016) and locality sensitive hashing Ji et al. (2012). Consequently, in our implementation and all experiments, we first generate a random JL matrix $S$ with i.i.d. Gaussian entries and then orthogonalize its rows using QR decomposition. We then use this orthogonalized matrix in our QJL quantizer, as described in Algorithm 1.

### 4.2 ABLATION STUDY

Here, we perform an ablation study on the relative distortion of the attention scores in one attention layer after applying QJL on key embeddings. The distortion for various layers of the Llama2-7B model is plotted against the number of bits per token and embedding channel $m/d$, where $d = 128$ is the embedding dimension, as shown in Figure 3. Our theoretical result from Theorem 3.6 suggests that $m \sim 1/\varepsilon^2$ which aligns with our observations in Figure 3. An interesting observation is that the first layer has a much higher distortion compared to all other layers, suggesting that the first layer is more challenging to quantize and requires a higher number of bits per FPN. This finding is noteworthy and indicates the need for tailored quantization strategies for different layers. This is consistent with the outlier distribution depicted in Figure 2, where the first layer appears distinct from the others.

### 4.3 END-TO-END TEXT GENERATION

Next we benchmark our method on LongBench Bai et al. (2023), a benchmark of long-range context on various tasks. We choose the base model as longchat-7b-v1.5-32k Li et al. (2023) (fine-tuned Llama-2 with 7B parameter with 16,384 context length) and apply following quantization methods to this model; KIVI Liu et al. (2024b), KVQuant Yue et al. (2024) and our proposed quantization via QJL. Each floating-point number (FPN) in the base model is represented by 16 bits, and we choose proper hyper-parameters of KIVI and QJL so that their bits per FPN become 3. For KVQuant, we follow the default setting which holds its bits per FPN as 4.3.

**QJL evaluation on Llama2 model and LongBench dataset.** We benchmarked QJL on Long-Bench Bai et al. (2023), a suite of tasks designed to evaluate performance with long-range contexts. We choose the base model as longchat-7b-v1.5-32k Li et al. (2023) (fine-tuned Llama-2 with 7B

| Methods | Bits | Datasets from LongBench Bai et al. (2023) | | | | | |
|---|---|---|---|---|---|---|---|
| | | NarrativeQA | Qasper | MultiQA-en | MultifQA-zh | HotpotQA | 2WikiMultiQA |
| FP16 (baseline) | 16 | 20.79 | 29.42 | 42.83 | 34.33 | 33.05 | 24.14 |
| KIVI Liu et al. (2024b) | 3 | **20.96** | **29.01** | 40.93 | **34.75** | 32.79 | 23.01 |
| QJL (ours) | 3 | 20.67 | 28.48 | **40.94** | 29.71 | **35.62** | **23.60** |
| KVQuant Hooper et al. (2024) | 4.3 | 20.14 | 28.77 | **44.22** | **34.44** | 34.06 | **23.05** |
| QJL (ours) | 4.3 | **20.72** | **30.02** | 41.18 | 31.73 | **34.22** | 22.63 |
| KIVI Liu et al. (2024b) | 5 | 20.49 | 28.90 | **43.24** | **34.66** | 33.07 | **24.86** |
| QJL (ours) | 5 | **21.09** | **29.11** | 41.58 | 31.86 | **35.65** | 24.61 |

Table 1: Evaluation of various quantization methods and different bits per floating-point number (FPN) on long-context question-answering datasets from LongBench (F1 scores).

parameter with 16,384 context length) and apply following quantization methods to this model; KIVI Liu et al. (2024b), KVQuant Hooper et al. (2024) and our proposed QJL. Each floating-point number (FPN) in the base model is represented by 16 bits. We chose several hyperparameters for QJL to match the bits per FPN of the competing methods KVQuant and KIVI. There are two versions of KIVI, with bits per FPN of 3 and 5, respectively. For KVQuant, the default setting results in 4.3 bits per FPN. To validate the quality of those quantized models, we benchmark them on 6 question-answer datasets from LongBench, and we set the maximum sequence length to 31,500. We follow the same approach of prompting and evaluating to evaluate the prediction of the model from the original repository. Table 1 summarizes the results. Our proposed QJL achieves the highest F1 score within the quantization methods for `NarrativeQA`, `Qasper` and `2WikiMultiQA`.

**Experiments with Llama3 and Llama2 models.** We additionally test our method on datasets `Lambada-OpenAI`, `HellaSwag`, `PIQA`, `MathQA`, and `MMLU`, which have shorter sequence lengths. We benchmark our method using LM-eval Gao et al. (2023) framework to ensure a thorough evaluation across various metrics. We evaluate quantization methods with accuracy across Llama-2-7B Touvron et al. (2023) and Llama-3-8B Llama3 (2024) models. Note that KIVI only supports a half-precision floating point, whereas our method can be used for any precision format type. This makes it unable to run KIVI on the Llama-3 model.

As we observe, QJL can significantly reduce memory usage by utilizing only 3 bits per FPN, compared to the 16 bits per FPN in the baseline, achieving around an 81% reduction in memory. We observe that this efficiency does not compromise performance significantly. Across all datasets, our method's accuracy is generally comparable to the baseline, with slight variations. In Table 2, our QJL on the Llama-3-8B performs on average about slightly better than the baseline across all datasets.

| Models | Methods | Bits | Datasets from LM-eval Gao et al. (2023) | | | | |
|---|---|---|---|---|---|---|---|
| | | | Lambada-OpenAI | HellaSwag | PIQA | MathQA | MMLU |
| | FP16 (baseline) | 16 | 73.90 | 57.18 | 78.07 | 28.11 | 41.85 |
| Llama-2-7B | KIVI Liu et al. (2024b) | 3 | 73.88 | 57.13 | 78.07 | 28.11 | 41.81 |
| | QJL (ours) | 3 | 73.88 | 57.14 | 78.07 | 28.17 | 41.78 |
| Llama-3-8B | BF16 (baseline) | 16 | 75.59 | 60.17 | 79.65 | 40.64 | 62.09 |
| | QJL (ours) | 3 | 75.61 | 60.13 | 79.87 | 40.60 | 62.12 |

Table 2: Evaluation (accuracy) of various quantization methods on regular length datasets from LM-eval Gao et al. (2023). These comparisons are not typically based on long-context length; however, even in these cases, our QJL with 3 bits per FPN performs comparably to the baseline with 16 bits per FPN.

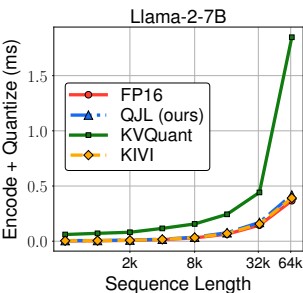 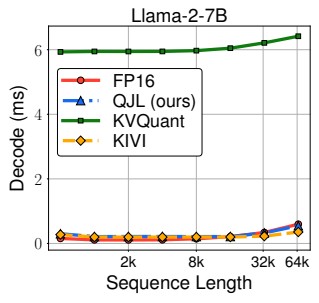 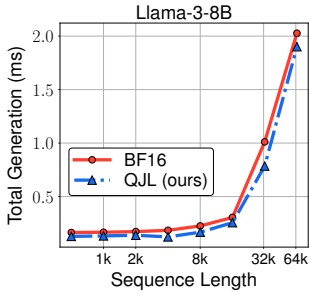

(a) Prompt encoding (Llama2)    (b) Token generation (Llama2)    (c) Encode and generate (Llama3)

Figure 4: Wall-clock time (ms) to encode a prompt and quantize the KV cache (left), generate 128 tokens for llama2 model (middle), and generate 64 tokens for llama3 model (right) using different quantization methods in a single attention layer model. The input sequence length varies from 1k to 64k. Both KIVI and QJL (ours) with 3 bits per FPN show faster decoding time than the baseline. However, KVQuant is significantly slower during both quantizing and decoding phases. QJL is the only method that can quantize Llama3, as our kernels support grouped query attention and BF16 data type. We observe the same speed for Llama3 as the exact method for generation. Note that our memory usage is at least 5-fold less than the exact method and can support all data types.

**Runtime and Peak-Memory Evaluations.**    To evaluate the runtime and memory consumption of QJL we additionally report runtimes of: **(1)** prompt encoding, **(2)** KV cache quantization, and **(3)** decoding (token generation) in a single attention layer as well as the **(4)** peak memory consumption during prompt encoding and decoding. Figure 4 shows the wall-clock time to encode a prompt and quantize the KV cache, generate 128 tokens for Llama2 model, and generate 64 tokens for Llama3 model using different quantization methods in a single attention layer of these models. Note that QJL is the only method that can quantize Llama3, as our kernels support grouped query attention and BF16 data type. we observe the same speed for Llama3 as the exact method for generation. The input sequence lengths vary between 1k to 128k. As shown in Figure 4, KVQuant runs slower than other methods during both prompt encoding and decoding phases, as it requires a huge amount of preprocessing which leads to slow runtime. On the other hand, both KIVI and our QJL with 3 bits per FPN show marginal runtime overhead compared to the exact baseline during prompting but reduce KV cache memory usage by at least a factor of 5.

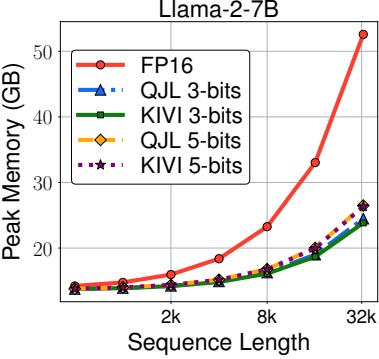

Figure 5: Peak memory usage for encoding the prompt and generating 128 tokens with Llama2, comparing various KV cache quantization methods to the exact model without quantization.

Next, we compare the peak memory consumption of various KV cache quantization methods applied to the Llama2 model for encoding prompts of different lengths and generating 128 new tokens, as shown in Figure 5. Both QJL and KIVI quantize the KV cache to 3 or 5 bits per FPN. However, peak memory consumption also includes the memory required to store model parameters. Even considering total memory consumption, we observe an over two-fold reduction in peak memory usage. We did not include KVQuant in the peak memory study as this method was extremely slow and running it repeatedly for different sequence lengths takes a very long time.

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
