# OpenReview forum: "QJL: 1-Bit Quantized JL Transform for KV Cache Quantization with Zero Overhead"
_ICLR.cc/2025/Conference — ICLR 2025 Conference Withdrawn Submission_

### Official Review · Reviewer_mqJ2 · 2024-10-31

**Soundness:** 2
**Presentation:** 3
**Contribution:** 2
**Rating:** 3
**Confidence:** 5

**Summary:**

The paper introduces QJL, a novel quantization method aimed at compressing the Key-Value (KV) cache memory in large language models (LLMs). QJL combines a Johnson-Lindenstrauss (JL) transform with sign-bit quantization, eliminating the need for additional quantization constants and thus reducing memory overhead. The authors also propose an asymmetric inner product estimator: by applying QJL to one vector and a standard JL transform (without quantization) to the other, they achieve unbiased, low-distortion inner product estimates. Experimental results demonstrate that QJL achieves comparable accuracy at various quantization bit levels while offering faster computational speed.

**Strengths:**

S1: This paper clearly describes the existing problem of KV cache compression.

S2: The paper effectively combines the KV cache compression problem with the Johnson-Lindenstrauss (JL) transform, leveraging the mathematical principles of the JL transform to compress the K cache.

S3: The paper theoretically derives the unbiased estimation and bounded distortion of the QJL algorithm.

**Weaknesses:**

W1: As shown in Theorem 3.6, the effectiveness of the QJL algorithm is proportional to the norms of the embeddings, necessitating a preprocessing step for the k cache in practical applications. In the paper, the authors illustrate this with a key cache plot in Figure 2, which leads to case-by-case handling and reduces usability.

W2: The authors only experimented with select quantization bit levels (3, 4.3, and 5 in Table 1; 3 in Table 2), leaving the experiments somewhat insufficient.

W3: The existing results in Tables 1 and 2 show no clear superiority over comparative algorithms, only comparable performance (slightly better on some tasks, slightly worse on others).

W4: The theoretical analysis could be enhanced with a discussion on time complexity.

**Questions:**

Q1: Regarding W1, please consider finding a method to automate end-to-end quantization of the k cache, eliminating the need for case-by-case manual adjustment.

Q2: Regarding W2, please add experimental results for higher bit levels in Tables 1 and 2, such as 8-bit for Table 1 and 4-bit, 5-bit, and 8-bit for Table 2.

Q3: Regarding W3, please explain how the current experimental results demonstrate the advantages of the QJL algorithm.

Q4: Regarding W4, please analyze the time complexity of the QJL algorithm in the theoretical section.

**Details Of Ethics Concerns:**

I believe this paper does not require an ethics review.

---

### Official Review · Reviewer_fk6D · 2024-11-03

**Soundness:** 2
**Presentation:** 3
**Contribution:** 2
**Rating:** 5
**Confidence:** 4

**Summary:**

This paper proposes a new KV cache compress method that first performs the johnson-Lindenstrauss (JL) transform to reduce outliers in KV cache. Then quantize the KV cache for LLM inference efficiency.

**Strengths:**

1. Well-theoretical analysis of JL transform and clear algorithm description.

2. Ablation study of long context dataset and system evaluation results.

**Weaknesses:**

### 1. Lack of Reasoning Datasets Evaluation
This comment highlights a legitimate concern but does not specify how the absence impacts the overall validity or generalization of the work. Additionally, it lacks any guidance or reference on the types of baselines expected.

### 2. Ambiguous Figures and illustration.
The comment points out an issue but could be clearer about the nature of the ambiguity. Mentioning only one figure limits the scope and leaves the authors guessing which visual improvements are needed. Also, the bit shown in the table is not well illustrated. KIVI actually supports 2-bit and 4-bit versions while in the paper it claims that KIVI only supports 3 and 5-bit versions.

### 3. Lack of Maximum Throughput Evaluation
The review raises an important point about throughput evaluation but does not discuss how this impacts the paper’s conclusions or how to address the gap meaningfully.

### 4. Ambiguous Illustration of Algorithm Workflow
The comment points out ambiguity but is vague about the specific aspects of the workflow that are unclear.(See details at questions)

**Questions:**

1.The title claims that the algorithm is 1-bit quantized; however, in the main text, it states that the KV cache is quantized to 3 bits. Could the authors clarify which representation is correct or explain if these refer to different bit representations?

2.The statement that "zero point and scale factor" add 1-2 bits per quantized value appears to be inaccurate. For KIVI, assuming a group size of 64, the memory overhead from scale and zero point is around 3%, which is approximately equivalent to a half-bit increase. Could the authors address this discrepancy?

3.The term "bit per floating point" in the paper is unclear. Does it refer to the bit width of the compressed KV cache or a metric for comparing the bit efficiency of different compression methods?

4.Following up on question 3, my understanding is that current NVIDIA GPUs do not natively support 3-bit quantization or an int3 data format. How is 3-bit quantization implemented in your approach, and does it introduce any additional latency?

5.In Section 4.3, the authors claim that KIVI does not support LLaMA3-8B. However, it seems possible to transfer the weights and activations of LLaMA-3-8B to half-precision (hf16) or use simulated compression for KIVI, which should yield similar accuracy. Could the authors comment on this?

6.Could the authors provide results on reasoning datasets like GSM8K or Math using advanced models such as Phi3 or Qwen2, comparing KIVI, QJL, and the fp16 baseline?

7.The system evaluation is incomplete. Could the authors provide throughput results for QJL on a single GPU, showing batch size versus maximum throughput? Since one of the key benefits of KV cache compression is improving the maximum throughput of LLM inference, this data would be crucial.

8.Related works, such as Quarot, use Hadamard transforms to mitigate outliers. How does the JL transform in your approach compare in terms of both accuracy and computational efficiency?

---

### Official Review · Reviewer_A7GW · 2024-11-04

**Soundness:** 1
**Presentation:** 2
**Contribution:** 2
**Rating:** 3
**Confidence:** 4

**Summary:**

This paper proposes QJL, a method of KV cache quantization for improving the memory-efficiency and throughput of LLM inference. The authors identify a problem in existing methods: there is significant memory overhead for storing quantization constatns. The authors propose QJL to eliminate this memory overhead, by leveraging Johnson-Lidenstrauss transform and sign-bit quantization. Empirical evaluations demonstrate competitive accuracy and inference efficiency against existing methods.

**Strengths:**

1. The problem studied is an important one.
2. The paper provides theoretical justifications for the proposed method.

**Weaknesses:**

1. Some claims in the paper may be overclaims. On line 14, the authors state that "traditional quantization methods face significant memory overhead due to the need to store quantization constants." On line 116, the authors claim that "QJL sketch can quantize vectors with zero overhead because it does not require grouping the data and storing quantization constants (zeros and scales) per group." However, line 175 makes clear that QJL uses "1-bit JL transform". Hence QJL has the non-zero overhead of 1 bit, which is the same overhead as KIVI for storing the quantization constants. Furthermore, QJL is only applied to keys and not values, so the overhead of value quantization is the same as existing methods.
2. The improvement over the baselines are marginal. From Table 1 and 2, the improvements over the baselines KIVI and KVQuant are mostly marginal. Furthermore, on certain datasets, QJL are considerably worse than the baselines by a few points (Table 1).
3. The "Conclusion" and "Related Works" sections are missing from the paper. The paper ends abruptly after experiments, with no conclusion or related works. This makes the paper incomplete.
4. The advantages and benefits of the method over existing methods are not clear. QJL has the same quantization overhead of 1-bit as KIVI, so it has the same memory efficiency. QJL is performing worse (in average accuracy) in all 3 comparisons in Table 1 against existing methods. Moreover, QJL does not offer better inference efficiency against KIVI. Can the authors please clarify the advantages of QJL over existing methods?

**Questions:**

1. QJL is only applied to keys and not values. Can QJL be applied to values as well? What are the impact of QJL for quantizing values on model accuracy?

---

### Official Review · Reviewer_tLHF · 2024-11-04

**Soundness:** 2
**Presentation:** 2
**Contribution:** 1
**Rating:** 3
**Confidence:** 5

**Summary:**

This paper introduces a novel approach to compressing key-value (KV) caches in large language models by utilizing a Quantized Johnson-Lindenstrauss (QJL) transform. The proposed method combines random projections and norm storage to approximate inner products, allowing for substantial memory reduction without significantly impacting model accuracy. By only storing the sign of the projected vectors and their norms, the method maintains the fidelity of inner product calculations essential for attention mechanisms. The paper provides rigorous mathematical analysis to validate the approximation accuracy and includes experimental results demonstrating its effectiveness in reducing memory usage while preserving performance in practical applications.

**Strengths:**

1.The approach proposed in this paper is innovative, utilizing random projections and norm storage during the compression process to ensure approximate preservation of inner product results. This provides a novel perspective for KV cache compression, offering a new direction in model optimization.

2.The mathematical proofs are rigorous and reliable, providing a solid estimation of the error bounds. This careful analysis enhances the credibility of the method and supports its practical application.

3.The paper clearly articulates the method's rationale and experimental setup, making it easy for readers to understand the approach and reproduce the experiments. The clarity in both the methodology and results presentation makes this work accessible and valuable to the research community.

**Weaknesses:**

1.The experimental results are inadequate and do not demonstrate that the method proposed in this paper is superior to existing methods.

2.The experiments are insufficient, failing to adequately showcase the performance of the proposed method compared to existing methods across different compression ratios and datasets.

3.The assumption of not storing the random matrix is unreasonable, as the proof relies on the assumption that the random matrices before and after are the same; the justification for not storing the random matrix is unconvincing.

4.There is no examination of the method's robustness under different random seeds.

**Questions:**

1.It is recommended to measure experimental results across different datasets and compression ratios.

2.Please enhance the discussion regarding the random matrix, considering the space occupied by the random matrix or providing proof that not storing the random matrix yields comparable results.

3.Use different random seeds to compute the mean and variance of the results to demonstrate robustness.

---

### Note · Authors · 2024-11-24

I have read and agree with the venue's withdrawal policy on behalf of myself and my co-authors.